# Effect of *CXCL17* on Subcutaneous Preadipocytes Proliferation in Goats

**DOI:** 10.3390/ani13111757

**Published:** 2023-05-25

**Authors:** Guangyu Lu, Xiaotong Ma, Fei Wang, Dingshuang Chen, Yaqiu Lin, Youli Wang, Wei Liu, Yanyan Li

**Affiliations:** 1College of Animal & Veterinary Sciences, Southwest Minzu University, Chengdu 610041, China; 2Key Laboratory of Qinghai—Tibetan Plateau Animal Genetic Resource Reservation and Utilization, Ministry of Education, Southwest Minzu University, Chengdu 610041, China

**Keywords:** goat, *CXCL17*, gene cloning, biological properties, adipocyte, proliferation

## Abstract

**Simple Summary:**

There are many existing studies related to goat adipose deposition. However, no studies have reported the role of the *CXCL17* gene on adipose deposition in goats before. The aim of this study was to explore the effect of *CXCL17* on goat subcutaneous preadipocytes proliferation and provide potential targets for goat-directed molecular breeding. The results showed that the exogenous expression of *CXCL17* at different doses had different regulatory effects on the proliferation of subcutaneous preadipocytes of goats.

**Abstract:**

The presence or absence of subcutaneous adipose accumulation will affect the energy storage, insulation resistance and metabolism of animals. Proliferation and differentiation of preadipocytes play a significant role in lipid deposition. The objective of this study was to clone the goat *CXCL17* gene and investigate its potential functions on goat subcutaneous preadipocytes’ proliferation by gaining or losing function in vitro. The goat *CXCL17* gene was cloned by Reverse Transcription-Polymerase Chain Reaction (RT-PCR) and bioinformatics analysis was performed. The expression of the *CXCL17* gene in the different goat tissues and adipocytes at different differentiation stages was detected by real-time fluorescence quantitative PCR (qPCR). The results showed that the cloned sequence of goat *CXCL17* gene is 728 bp and the CDS region is 357 bp, encoding 118 amino acids. *CXCL17* protein is located in nucleus, cytoplasm, mitochondria and extracellular matrix. Tissue-expression profiles revealed that *CXCL17* expressed in all of the examined tissues. In visceral tissues, the highest expression level was found in lung (*p* < 0.01); in muscle tissues, the highest *CXCL17* expression level was found in the longissimus dorsi (*p* < 0.01) and in adipose tissues, the highest expression level was found in subcutaneous adipose (*p* <0.01). Compared with those cells before differentiation, *CXCL17* expression levels upregulated at 48 h (*p* < 0.01), 72 h (*p* < 0.01), 120 h (*p* < 0.01) and downregulated at 96 h (*p* < 0.01). Furthermore, the results of crystal violet staining and semi-quantitative assay showed that transfection with 1 μg *CXCL17* expression plasmid reduced the cell numbers in vitro. Meanwhile, the expression of *CCND1* was significantly decreased. A similar consequence happened after interfering with *CXCL17* expression. However, plasmid transfected with 2 μg pEGFPN1-*CXCL17* increased the number of cells in vitro. These results suggest that *CXCL17* is involved in the proliferation of goat subcutaneous preadipocytes.

## 1. Introduction

In humans, there are many diseases related to adipose tissue [1,2,3]. Subcutaneous fat (SAT), intramuscular fat (IMF) and visceral fat (VAT) constitute the mammalian adipose tissue. For meat-used animals, IMF and SAT are highly valued because they are important factors affecting meat quality [4]. Subcutaneous adipose tissue plays an important role in the stabilization of lipid metabolism balance and is the main site of lipid deposition [5]. lipid deposition can seriously affect their meat quality, meat flavor and carcass lean meat percentage [6,7]. Therefore, it is important to explore the metabolic characteristics and regulatory mechanism of subcutaneous adipose for the meat value of meat-used animals.

Chemokines (also known as chemotaxis hormone) is a small-molecule cytokine family of proteins that can collect leukocytes to clear foreign bodies such as foreign pathogens and plays an important role in inflammatory response [8]. By classifying the relative positions of the cysteine residues at the amino terminal (N end), the chemokines can be divided into four subgroups: CXC (major), CC (major), XC (minor) and CX3C (minor) [9]. According to the triplet glutamate-to-leucine-arginine sequence (ELR) before the first cysteine, the CXC subgroup can be further divided into ELR-positive (ELR +) and -negative (ELR −). CXC motif chemokine ligand 17 (*CXCL17*) has obvious angiogenesis-promoting effect and belongs to the ELR + class of CXC subgroup [10]. The *CXCL17* gene localized on human chromosome 19q13.2 was found in 2006 and consisted of four exons. The precursor protein (DMC) contains 119 amino acid residues, six of which are cysteines. Human *CXCL17* contains a signal peptide consisting of 22 amino acids [11].

*CXCL17* can recruit monocytes, macrophages and mature and immature dendritic cells (DC), which is of great significance for the development of multiple cancer types [12,13,14,15,16,17,18]. Abnormal expression of *CXCL17* in tumor cells can accumulate immature myeloid cells and promote cancer development by promoting vascular proliferation [19]. In addition, *CXCL17* is an antibacterial mucosal chemokine, which can not only exert antibacterial activity through a peptide-mediated way to disrupt bacterial membrane, but also reduce the expression of type I collagen by upregulating the expression of matrix metalloproteinase 1 (MMP1) and miR-29 in fibroblasts, thus participating in the occurrence of systemic sclerosis (SSc) [20,21]. Moreover, *CXCL17*/*CXCR8* signaling pathway plays an important role in mucosal immunity in the female genital tract [22]. *CXCL17* is highly expressed in salivary glands and lacrimal glands of patients with primary Sjogren’s syndrome [23]. Since *CXCL17* has previously been reported to be involved in cell proliferation, preadipocytes proliferation promotes lipid accumulation. Goat lipid deposition plays an important role in meat quality improvement. Tissue expression detection result showed that *CXCL17* expression level was the highest in subcutaneous adipose tissue. Therefore, we speculated that it plays an important regulatory role in the process of subcutaneous fat deposition in goats.

The Jianzhou Daer goat is the second breed of goat bred independently in China and has high meat value. Subcutaneous adipose deposition can reduce the proportion of lean meat in the carcass and has a great impact on meat quality [24]. In this study, we cloned the *CXCL17* coding sequence and analyzed its biological characteristics by the bioinformatics method. Real-time PCR was used to analyze the expression level of *CXCL17* gene in different tissues and subcutaneous preadipocytes at different differentiation induction times. Moreover, we analyzed its effect on cell proliferation using gain-of-function and loss-of-function experiments. Therefore, this study aims to characterize the potential functions of *CXCL17* on the proliferation of goat subcutaneous preadipocytes and provide a new theoretical basis for further study of lipid deposition in goat adipocytes.

## 2. Materials and Methods

### 2.1. Test Animals and Samples Collection

The test animals are one-year-old healthy Jianzhou Daer goats (*n* = 3) that were purchased from Sichuan Tiandi Yang Biological Engineering Co., Ltd., (Chengdu, China) and sampled immediately after slaughter. The samples included heart, liver, spleen, lung, kidney, large intestine, small intestine, rumen, longissimus dorsi, vastus, gluteus, abdominal adipose, subcutaneous adipose, mesenteric adipose, pericardial adipose, intermuscular adipose and visceral adipose tissue. The Institutional Animal Care and Use Committee of Southwest Minzu University (Chengdu, China) permitted this research project, and all animal experiments in this study were in line with animal ethical treatment.

### 2.2. Total RNA Extraction and Reverse Transcription

Total RNA was extracted from each tissue and cells by Trizol (TaKara, Dalian, China). Then, 1 μg RNA was reverse transcribed according to the instructions of reverse transcription kit (Thermo, Waltham, MA, USA) and stored at −20 °C.

### 2.3. Cloning of Coding Region of Goat CXCL17

For gene cloning, primers were designed according to the mRNA sequence of bovine *CXCL17* gene in GenBank (accession number: NM_00101014862.2), and the sequence of primers were as follows: sense: 5′ CGCCCTCTAATGAGAATGCT 3′ antisense: 5′ AGCGTAAGGCTGTGTGAAGG 3′. Complementary DNA (cDNA) of goat cardiac tissue was used for PCR amplification. As described in the previous article in our laboratory, the positive bacterial solution was sent to Chengdu Qingke Biotechnology Co., Ltd. (Chengdu, China) for verification and sequencing [25].

### 2.4. Bioinformatics Analysis of Goat CXCL17 Coding Region

The analysis software and online tools of goat *CXCL17* gene refer to Table 1, and the detailed analysis method is based on the relevant literature [26].

### 2.5. Induced Differentiation of Subcutaneous Preadipocytes

Primary culturing of subcutaneous goat preadipocytes was conducted according to the isolation and culture methods established in the laboratory [27]. Goat subcutaneous preadipocytes cultured to F3 were counted and inoculated into 12-well plates. When the cell confluence reached 80%, the complete medium was replaced with 50 μmol/L oleic acid induction solution (Sigma, Tokyo, Japan), and the fluid was changed every 2 d. Cells were collected at 0 h, 24 h, 48 h, 72 h, 96 h and 120 h.

### 2.6. Real-Time Fluorescence Quantitative PCR (qPCR)

According to the instructions of SYBR ® Premix Ex Taq ^TM^ (2×) kit (TaKara, Kusatsu, Shiga, Japan), the expression levels of *CXCL17* and proliferation marker genes were determined using *UXT* as an internal reference gene. *CXCL17* quantitative primers and expression vector construction primers were designed by cloned *CXCL17* gene sequence, and other primers were designed by goat sequences in NCBI. Primer information is listed in Table 2.

### 2.7. Construction of Overexpression Plasmid and Synthesis of siRNA

For construction of overexpression plasmid, the sequence of goat *CXCL17* gene (GenBank accession number: ON930036) was amplified by PCR. After detection by 1% agarose gel electrophoresis, the PCR product was purified and recycled. Then the PCR products and the pEGFP-N1 plasmid were digested by restriction enzymes (*Eco*R Ⅰ; TaKara, Dalian, China; *Kpn* Ⅰ; TaKara, Dalian, China) and ligated by T4 DNA Ligase (TaKara, Dalian, China) at 16 °C overnight. Transforming and coating plates were used. Then colonies were selected and identified by PCR, and sent to Chengdu Qingke Biotechnology Co., Ltd. (Qingke, Chengdu, China) for sequencing. After successful sequencing, we used plasmid extraction kits (TIANGEN, Beijing, China) to extract the plasmid and stored it at 4 °C.

For siRNAs, the siRNAs were synthesized by GenePharma (GenePharma, Shanghai, China). The sequence of siRNAS were as follows:

si-*CXCL17*: sense: 5′-CCAAGAAUGUGAGUGCCAATT-3′;

antisense: 5′-UUGGCACUCACAUUCUUGGTT-3′;

si-NC: sense: 5′-UUCUCCGAACGUGUCACGUTT-3′;

antisense: 5′-ACGUGACACGUUCGGAGAATT-3′.

### 2.8. Transfection

When goat subcutaneous preadipocytes were cultured to F3 generation and reached about 80% confluence, cells were planted into 6-well plates or 96-well plates. The control group was transfected with pEGFP-N1 (Vector) or siRNA (NC), and the experimental group was transfected with pEGFP-*CXCL17* or siRNA-*CXCL17*, respectively.

Transfection system for overexpression (6-well plates): 1 μg/2 μg pEGFP-*CXCL17*/pEGFP-N1, 400 μL Opti-MEM (Gibco, Calabasas, CA, USA), 6 μL Hieff Trans^TM^ Liposomal Transfection Reagent (Yeasen, Shanghai, China).

Transfection system for interference (6-well plates): 6 μL (20 μM) siRNA-*CXCL17*/siRNA (NC), 400 μL Opti-MEM (Gibco, Calabasas, CA, USA), 6 μL Hieff Trans^TM^ Liposomal Transfection Reagent (Yeasen, Shanghai, China).

Transfection system for overexpression (96-well plates): 0.1 μg/0.2 μg pEGFP-*CXCL17*/pEGFP-N1, 20 μL Opti-MEM (Gibco, Calabasas, CA, USA), 3 μL Hieff Trans^TM^ Liposomal Transfection Reagent (Yeasen, Shanghai, China).

Transfection system for interference (96-well plates): 0.3 μL (1 μM) siRNA-*CXCL17*/siRNA (NC), 20 μL Opti-MEM (Gibco, Calabasas, CA, USA), 3 μL Hieff Trans^TM^ Liposomal Transfection Reagent (Yeasen, Shanghai, China).

### 2.9. MTT Assay

MTT experiments were performed to detect the cells’ viability. We planted goat subcutaneous preadipocytes in 96-well plates with a density of 3 × 10^3^ per well and cultured them according to the methods of previous articles in our laboratory [28]. Adding 10 µL MTT reagent (solarbio, Beijing, China) into each well of 96-well plates away from light, cells were cultured at 37 °C and 5% CO_2_ for 4 h. The absorbance at 490 nm was measured by using an enzyme-labeled instrument.

For crystal violet staining, goat subcutaneous preadipocytes were seeded in 96-well plates with a density of 3 × 10^3^ per well. Culture media was discarded after 0, 24, 48 and 72 h of cell culture. Then goat subcutaneous preadipocytes were washed twice with PBS and fixed with 4% formaldehyde for 30 min. Next, 4% formaldehyde was discarded, and cells were washed twice with PBS. Then, 200 μL of 0.1% Crystal violet-solution was added into each well to stain for 20 min. After discarding Crystal violet solution, cells were washed twice with PBS. The images of crystal violet staining were captured by fluorescence microscopy (Olympus, Tokyo, Japan).

For semi-quantification, 200 μL of 20% glacial acetic acid was added to each well of the 96-well plate for 15 min. Then the optical density at 490 nm was detected using an enzyme-labeled instrument.

### 2.10. Statistical Analysis

Quantitative PCR data were analyzed using the 2^− ΔΔ Ct^ method, and the data were expressed as “mean ± standard deviation (Mean ± SD)”. The difference of the data was analyzed using the one-way ANOVA method, the multiple comparison t-test and the Student’s two-tailed t-test in GraphPad Prism 8.0 software. All experiments in this study including biological information analysis, qPCR, cell culture, MTT and semi-quantitative experiments were performed in triplicate. All experiments were statistically significant when *p* < 0.05. “*”: *p* < 0.05 indicates a significant difference and “**”: *p* < 0.01 indicates a very significant difference.

## 3. Results

### 3.1. Goat CXCL17 Gene Cloning

In order to study the biological function of *CXCL17* in goats, we cloned its coding sequence first. Goat heart tissue cDNA was used as the template to clone the *CXCL17* gene, resulting in a specific band consistent with the size of the expected target fragment (Figure 1A). After sequencing, the obtained sequence was 728 bp long and sequence analysis indicated that the *CXCL17* CDS region was 357 bp encoding 118 amino acids (Figure 1B). The GenBank accession number was ON930036.

### 3.2. Protein Structure and Amino Acid Composition Analysis of Goat CXCL17

The primary structure and physicochemical properties of goat *CXCL17* were analyzed by Ex PASy-ProParam. The protein molecular formula of *CXCL17* is C _592_ H _981_ N _193_ O _159_ S _10_, its relative molecular mass is 13,667.10 Da, the isoelectric point (pI) is 11.07, the total average of hydrophilic (GRAVY) is −0.582 and the instability index (II) is 57.27, indicating that the protein is an unstable basic hydrophilic protein. The content of various amino acids of the *CXCL17* protein is shown in Figure 2A, with the highest content of arginine (Arg) and leucine (Leu), and the least content of isoleucine (Ile) and tryptophan (Trp), respectively. The aliphatic index of *CXCL17* protein is 90.20 and the half-life is 30 h. The number of total residues of positive charge (Arg + Lys) and the total number of negative charge (Asp + Glu) are 24 and 8, respectively, indicating that the protein is positively charged.

According to the secondary structure prediction of goat *CXCL17* using SOPMA, it was found that 57 amino acids (48.31%) had the highest proportion of random curling, 46 amino acids (38.98%) could form an α helix, 12 amino acids could form an extension chain (10.17%) and 3 amino acids (2.54%) could form a β turn (Figure 2B). SWISS-MODEL software was used to predict the tertiary structure models of goat and other species’ proteins and the results showed that the spatial structure of goat *CXCL17* protein was extremely close to sheep and oryx dammah (Figure 2C).

### 3.3. Subcellular Localization, Signal Peptide Prediction, Protein Phosphorylation Site and Transmembrane Domain, Amino Acid Sequence Homology and Phylogenetic Tree Analysis

PSORT Ⅱ was used to predict subcellular localization of goat *CXCL17*. The result showed that 11.1% is located in the cytoplasmic matrix and nucleus, 33.3% in the mitochondria and 44.4% in the extracellular matrix (including cell membrane). SignalP-5.0 Server predicted signal peptide showed that *CXCL17* has a signal peptide (Figure 3A). The NetPhos 3.1-predicted result showed that there were 11 phosphorylation sites, 7 serine phosphorylation sites and 4 threonine phosphorylation sites (Figure 3B). The *CXCL17* transmembrane domain was predicted by TMHMM 2.0, which found that *CXCL17* has a transmembrane domain (Figure 3C).

The comparison results of amino acid sequence homology showed that goat *CXCL17* shared the highest similarity with ovis aries, and higher similarity with oryx dammah, bison, cervus elaphus and bos mutus (Figure 3D), indicating that *CXCL17* protein was highly conserved among different species. The phylogenetic tree was constructed by MEGA 5.0 according to the amino acid sequence homology between goat and other species. *CXCL17* of goat and ovis aries were in the same branch of the phylogenetic tree, indicating that goat *CXCL17* and ovis aries were closely related (Figure 3E).

### 3.4. Analysis of Tissue and Temporal Expression Profile of CXCL17 Gene in Goats

*UXT* was used as a reference gene to determine *CXCL17* expression level in various goat tissues, and qPCR detection results showed that *CXCL17* expressed in all the examined tissues [29]. In visceral tissues, *CXCL17* expression was highest in lung (*p* < 0.01), the next-highest expression tissue was rumen (*p* < 0.01), and low expression was observed in liver, small intestine, large intestine and spleen (Figure 4A). The muscle tissue expression profiles showed that *CXCL17* expressed in longissimus dorsi, gluteus (*p* < 0.01) and vastus (*p* < 0.01) with a trend from high to low (Figure 4A). In adipose tissue, the expression of *CXCL17* in subcutaneous adipose tissue was the highest (*p* < 0.01), followed by the higher expression in abdominal adipose (*p* < 0.01), and low expression in mesenteric adipose, intramuscular adipose and visceral adipose (Figure 4A). Because *CXCL17* gene expression was the highest in subcutaneous adipose, we then speculated that it may play a vital regulatory role in lipid deposition of goat subcutaneous preadipocytes. The expression of *CXCL17* gene during subcutaneous preadipocytes differentiation was determined by qPCR. The results showed that its expression was significantly upregulated at 48 h and 72 h (Figure 4B). Interestingly, the expression of *CXCL17* gene was reduced to about half of the original expression level at 96 h of induced differentiation, and the expression level was significantly upregulated at 120 h of induced differentiation (Figure 4B). The above results indicated that *CXCL17* may play different regulatory roles in different stages of differentiation of goat subcutaneous preadipocytes.

### 3.5. CXCL17 Expression Plasmid Construction

Preadipocytes proliferation is another factor affecting lipid deposition. To investigate the role of *CXCL17* in goat subcutaneous preadipocytes proliferation, *CXCL17* expression plasmid was constructed first. PCR amplifying results showed a correct strip location (Figure 5A). Primer information used to construct overexpressed vectors is shown in Table 2. The PCR product was connected to pEGFP-N1 vector after digestion, and the sequencing results were correct (Figure 5B). In order to clarify the localization of goat *CXCL17*, subcutaneous preadipocytes were transfected with exogenous pEGFP-*CXCL17* plasmid and took fluorescence photographs. In contrast to pEGFP-N1, pEGFP-*CXCL17* has many dispersed extracellular fluorescences, largely consistent with the subcellular localization results in 3.3 (Figure 5C). The results of overexpression efficiency showed that the expression level of *CXCL17* gene was nearly 2 400 000-fold upregulated when compared with the control group (Figure 5D). In conclusion, the pEGFP-*CXCL17* overexpression plasmid was constructed successfully.

### 3.6. Transfected Plasmid with Different Dose Showed Different Value-Added Effect

In order to study the effect of *CXCL17* on subcutaneous preadipocytes proliferation, we transfected 1 μg expression plasmid into goat subcutaneous preadipocytes. The results of qPCR showed that the transfection efficiency was about a 4 000-fold change, when compared to the negative control (Figure 6A). The detection results of proliferation marker genes expression showed that *CCND*1 were downregulated in the *CXCL17* overexpression group (Figure 6B).

To further verify the effect of *CXCL17* on the proliferation of subcutaneous preadipocytes in goats, we performed crystal violet staining and semi-quantitative experiments. The number of cells in the *CXCL17* group was significantly less than that in the control group at 72 h (Figure 6C). Furthermore, the semi-quantitative results were generally consistent with the crystal violet staining results (Figure 6D). We then examined the cell viability by MTT, and the result showed that *CXCL17* overexpression significantly inhibited cell proliferation after 24 h–72 h transfection (Figure 6E). Eventually, the above results suggested that *CXCL17* had a significant inhibitory effect on the proliferation of goat subcutaneous preadipocytes when transfected with a 1μg pEGFP-*CXCL17* overexpression plasmid.

To more accurately understand the effect of *CXCL17* overexpression on goat subcutaneous preadipocytes proliferation, we transfected 2 μg *CXCL17* plasmid into these cells. We first examined the transfection efficiency, and *CXCL17* was successfully overexpressed (Figure 7A). Subsequently, we detected the proliferation relative marker genes’ expression, and found the expression of *CDK2* and *CCND1* increased in the *CXCL17* overexpression group (*p* < 0.05) (Figure 7B).

Then we performed crystal violet staining to determine whether the exogenous overexpression of *CXCL17* has an effect on the cell numbers (Figure 7C). Meanwhile, from the semi-quantitative results in Figure 7D, there was significant difference in the cell numbers between the two groups at 72 h. The crystal violet staining result is generally in line with the semi-quantitative result, with significant differences in the cell number between the two groups at 72 h. We found that the cell viability in the overexpression group at 72 h (*p* < 0.01) was greater than that in the control group by MTT (Figure 7E). All the experimental results above showed that overexpression of high-dose *CXCL17* could promote goat subcutaneous preadipocytes proliferation.

### 3.7. Knockdown of CXCL17 Inhibits Goat Subcutaneous Preadipocytes Proliferation

To further confirm the effect of *CXCL17* on goat subcutaneous preadipocytes proliferation, three specific siRNA targets for *CXCL17* had been synthesized. As shown in Appendix A, the interference efficiency of siRNA-2 is the strongest, so we choose siRNA-2 (named si-*CXCL17*) for the following studies. We examined the transfection efficiency of siRNA-2 again to check the correctness of the results, and the qPCR results indicated the expression of *CXCL17* mRNA in si-*CXCL17* group was significantly lower than that in NC group (Figure 8A). Meanwhile, we examined the mRNA expression level of marker genes associated with cell proliferation and found the expression of marker genes *CCNE1* (*p* < 0.01) and *CDK2* (*p* < 0.05) in the si-*CXCL17* group was lower than the NC group (Figure 8B).

In addition to the above experiments, we performed crystal violet staining and found that the number of cells in the si-*CXCL17* group was less than that in the NC group at 72 h (Figure 8C). The results are in Figure 8D, and the absorbance of the si-*CXCL17* group was very much lower than the NC group at 72 h (*p* < 0.01). We also used MTT to analyze cell viability with the interference of *CXCL17* and found that cell viability in the interference group was significantly lower than in the NC group at 24 h–72 h (*p* < 0.01) (Figure 8E). Combined with disrupting all experiments, we speculated that decreased expression of *CXCL17* could inhibit subcutaneous preadipocytes proliferation in goats.

### 3.8. Analysis of CXCL17 Interaction Proteins

To study the mechanism of *CXCL17* inhibiting the proliferation of subcutaneous preadipocytes, the STRING interaction database was used to predict the potential interaction of proteins. The result indicated that *CXCL17* may interact with proteins such as GPR35, CCL17 and SNTN (Figure 9). Relevant studies showed that GPR35 knockdown will affect lipid accumulation in hepatocytes in mice [30]. Moreover, some studies reported that GPR35 is the receptor of *CXCL17* [31]. So, we hypothesized that *CXCL17* may affect subcutaneous adipose deposition by interacting with GPR35 to affect subcutaneous preadipocytes proliferation in goats.

## 4. Discussion

The human chemokine superfamily consists of 48 ligands and 20 receptors. They can regulate the migration of cells such as leukocytes and stem cells in vivo. Chemokines can promote the development of inflammatory response by clustering leukocytes in inflamed or injured tissues and by regulating leukocyte homeostasis and homing [32]. *CXCL17* is the last chemokine identified and characterized so far, which can promote angiogenesis, so it is considered to be a vascular endothelial growth factor (VEGF) coregulatory chemokine [33]. *CXCL17* has both homeostatic chemokines and inflammatory chemokines, which is the reason why it is considered as a “dual” chemokine. *CXCL17* has many functions, including antibacterial activity and antiviral function, and is highly chemotropic to bone marrow cells [34]. Multiple cases indicate a dual effect of *CXCL17* on human health, but the role of *CXCL17* in goats has not been reported before.

In this study, the mRNA sequence of cattle *CXCL17* gene was used as a template to design primers and clone goat coding sequence. Fluorescence localization revealed that transfected pEGFP-*CXCL17* had much extracellular fluorescence, consistent with the *CXCL17* precursor protein as a secreted protein [11]. The differential expression of goat *CXCL17* in various tissues was determined by qPCR technique. We found that goat *CXCL17* expressed in all the tissues examined. In visceral tissues, the expression of *CXCL17* in lung was the highest (*p* < 0.01), followed in rumen (*p* < 0.01). Related studies have shown that *CXCL17* plays an important role in the respiratory system-related diseases [17,20,35,36], which may explain the extremely high expression of *CXCL17* in goats in the lung. According to the relevant study, *CXCL17* not only plays an important role in the development and prognosis of gastric cancer, but is also related with some bacteria in the stomach [37,38,39]. So, we speculated that *CXCL17* may affect its digestive function by affecting the internal flora in the rumen of goats. In addition, the expression level of goat *CXCL17* in the liver is relatively high, and according to the results of related studies [14,39,40], we speculated that *CXCL17* may be related to the immune function of goat liver. In adipose tissues, the expression level of *CXCL17* was the highest in subcutaneous adipose and the lowest in pericardial adipose. Since our sample came from Jianzhou Daer goats, whether tissue expression detection experiments have the same results in other goat breeds is unknown and needs to be verified. Furthermore, our test results were a replication of three goats, and it is unknown whether increasing the sample size will affect the tissue expression profile of *CXCL17* in goats.

Meanwhile, in this study we found that *CXCL17* expression level increased in preadipocytes at 48 h, 72 h and 120 h. Consequently, we surmised *CXCL17* may play a role in promoting the adipose deposition of subcutaneous preadipocytes at these stages. Interestingly, the *CXCL17* expression level in preadipocytes was decreased at 96 h. It was speculated that the regulation of *CXCL17* in subcutaneous adipose deposition changed from positive to negative. In conclusion, these results suggest that *CXCL17* may play different regulatory roles in the different stages of adipose deposition of subcutaneous preadipocytes in goats. Relevant studies showed that *CXCL10* and *CXCL11* are potential predictive molecules for obesity onset and *CXCL1* expression level is related to adipocyte differentiation [41,42]. It was also reported that *CXCL-1* overexpression can reduce adipose accumulation in adipose tissue and, in the context of high glucose, the effect of adipocyte pyroptosis in different concentrations of *CXCL14* and the effect of *CXCL14* on adipocytes decreased first and then increased with time [43,44]. In addition, *CXCL17* and *CXCL14* are related in structure and function [45]. Based on these studies, we speculated that *CXCL17* may be related to lipid deposition in goat subcutaneous preadipocytes.

The expression vector transfected with 1 μg inhibited the proliferation of goat subcutaneous preadipocytes, while the expression vector transfected with 2 μg promoted the proliferation of goat subcutaneous preadipocytes, interfered with *CXCL17* expression and inhibited the proliferation of goat subcutaneous preadipocytes. These results indicated that different doses of *CXCL17* had different effects on the proliferation of subcutaneous preadipocytes of goats. Related studies reported that low-dose recombinant *CXCL17* reduced the number of spleen cells in mice, while high-dose recombinant *CXCL17* increased the number of spleen cells in mice [19]. Different doses of *CXCL17* showed different biological effects on the cells, which was consistent with the phenomenon of *CXCL17* in goat subcutaneous preadipocytes in this study. Therefore, it is highly likely that *CXCL17* needs to overcome the threshold before its inhibitory effect on the proliferation of subcutaneous preadipocytes of goats can be converted into promoting effect.

## 5. Conclusions

In conclusion, our data clarify *CXCL17* may have a modulatory effect on subcutaneous preadipocytes proliferation. Low dose of exogenous *CXCL17* expression inhibited the proliferation, while high dose of exogenous *CXCL17* expression promoted goat subcutaneous preadipocytes proliferation. Interference with *CXCL17* inhibited the proliferation of goat subcutaneous preadipocytes. These results indicate that *CXCL17* is a novel candidate gene for adipose deposition in goats. This study will provide a new theoretical basis for further study of lipid deposition in goat adipocytes and provide a basis for enriching the molecular regulatory network and mechanism of goat *CXCL17*.

## Figures and Tables

**Figure 1 animals-13-01757-f001:**
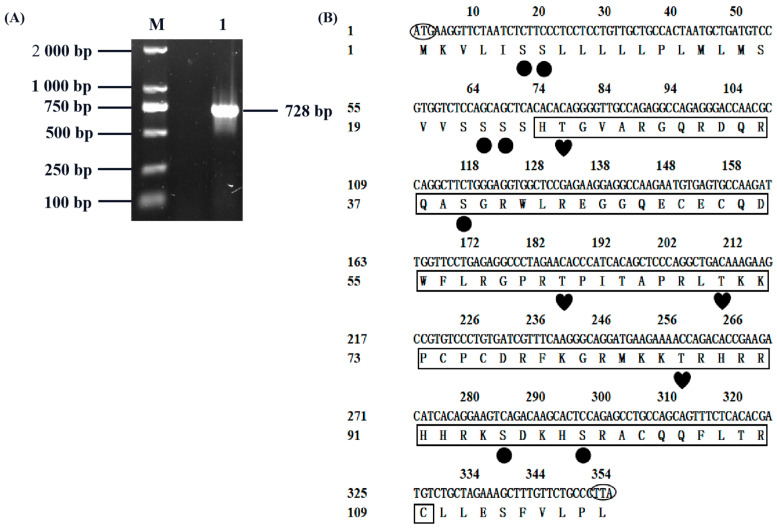
Cloning of *CXCL17* gene in goat. (**A**) Amplification of *CXCL17* coding region in goat; M: DL2000 DNA marker, 1: PCR products obtained in the amplification of the *CXCL17* coding region. (**B**) The sequences of open reading frame and deduced amino acid of *CXCL17* coding region in goat; Circles: Serine phosphorylation sites, Hearts: Threonine phosphorylation sites, Boxes: *CXCL17* functional domains, ATG: The start codon, TTA: The stop codon.

**Figure 2 animals-13-01757-f002:**
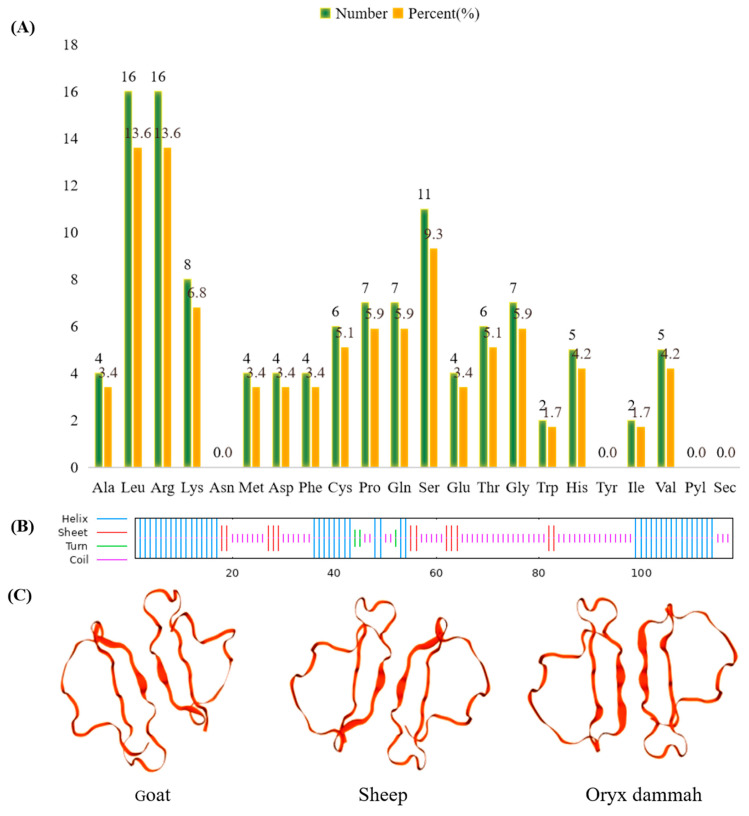
Sequence analysis of the *CXCL17* gene in goats. (**A**) Amino acid composition of goat *CXCL17* protein. (**B**) Prediction of secondary structure of goat *CXCL17* protein. (**C**) Prediction of *CXCL17* protein tertiary structure in goat and other species.

**Figure 3 animals-13-01757-f003:**
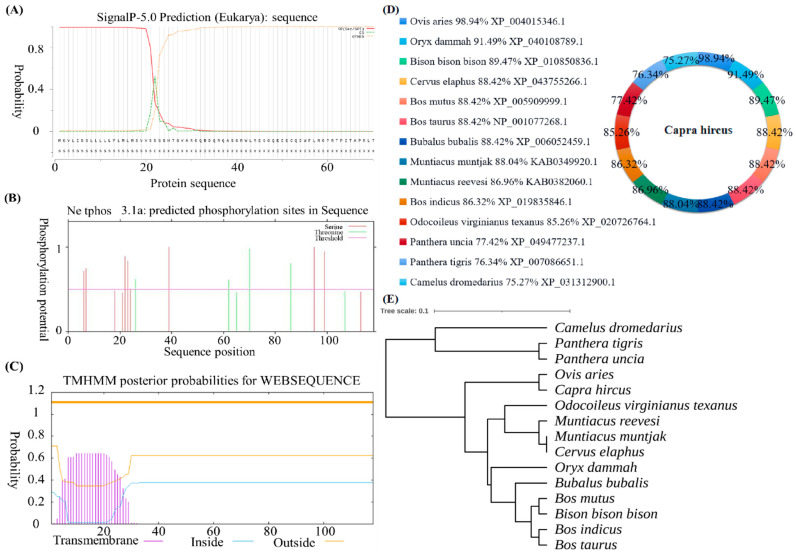
The subcellular localization, signal peptide prediction, protein phosphorylation site, transmembrane domain, amino acid sequence homology and phylogenetic tree analysis of goat *CXCL17*. (**A**) Prediction of *CXCL17* protein signal peptide. (**B**) Prediction of *CXCL17* protein phosphorylation site. (**C**) Prediction of *CXCL17* protein transmembrane helices structure. (**D**) Homology information of goat *CXCL17* amino acid sequence. (**E**) The phylogenetic tree of *CXCL17*.

**Figure 4 animals-13-01757-f004:**
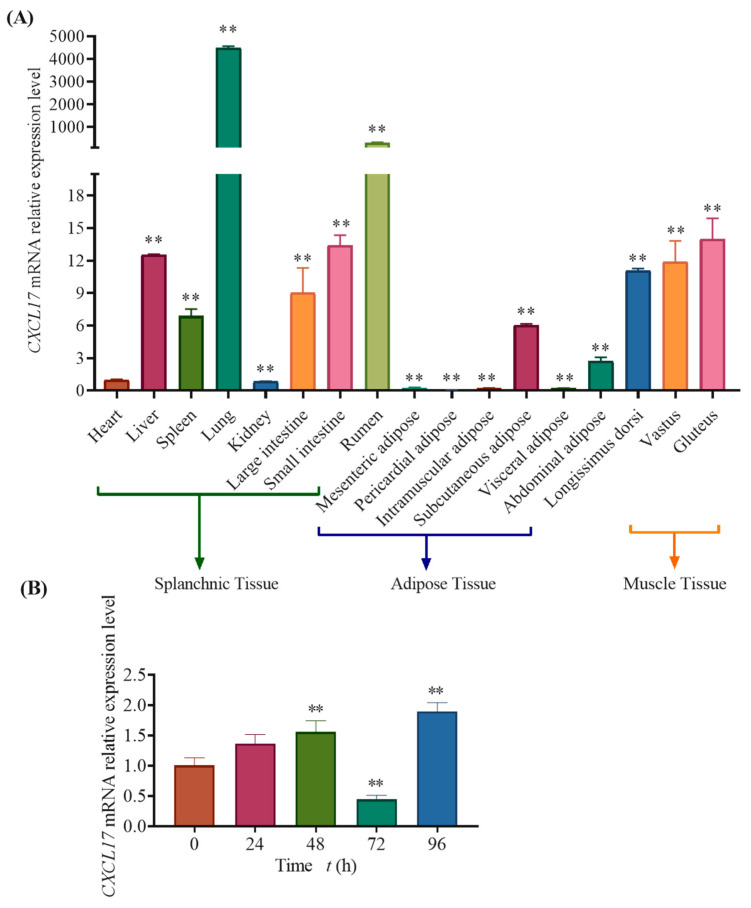
The relative expression of *CXCL17* gene in goat during subcutaneous preadipocytes differentiation and the expression profile of *CXCL17* gene in different tissues of goat. **: *p* < 0.01. (**A**) The expression profile of *CXCL17* in 14 tissues of goat. (**B**) The temporal expression profile of *CXCL17*. All experiments in our study were performed at least in triplicate.

**Figure 5 animals-13-01757-f005:**
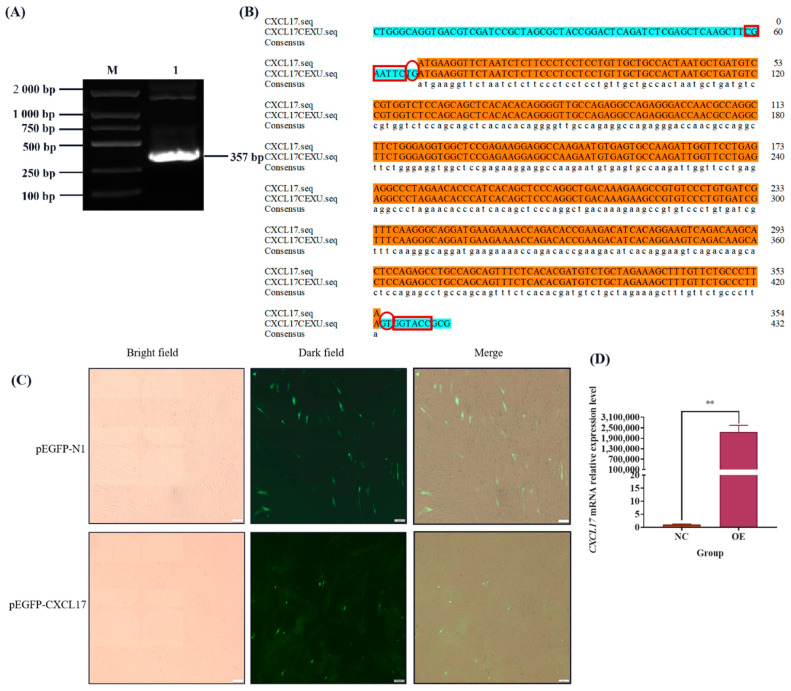
Over expression of *CXCL17* gene in goat. (**A**) Results of CDS amplification of goat *CXCL17* gene, M: DL2000 DNA marker, 1: Target strip in CDS region of goat *CXCL17* gene. (**B**) Comparison of sequencing results with CDS region of *CXCL17* gene, round boxes indicate the base to prevent the insertion of frameshift mutation, and square boxes indicate restriction sites. (**C**) Fluorescence localization of *CXCL17* protein in goat preadipocytes cultured in vitro (100×). (**D**) The overexpression efficiency of *CXCL17* gene, the transfection dose of expression plasmid was 3 μg, **: *p* < 0.01.

**Figure 6 animals-13-01757-f006:**
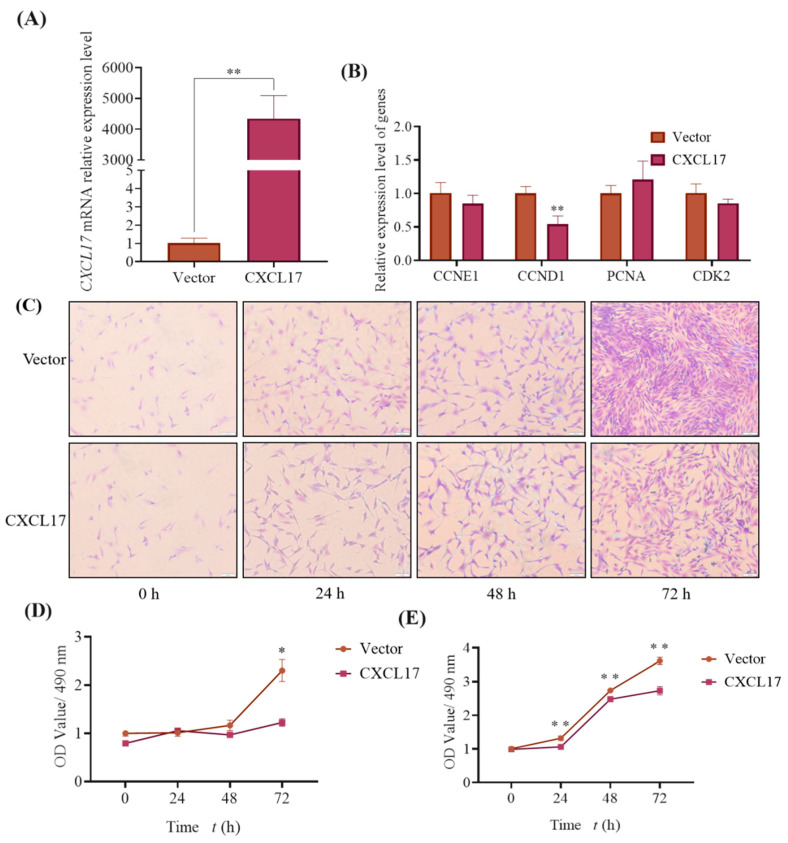
Overexpression of low-dose *CXCL17* inhibits goat subcutaneous preadipocytes proliferation. (**A**) Overexpression efficiency of *CXCL17* gene when transfected with 1 μg overexpression plasmid; **: *p* < 0.01. (**B**) Expression of genes associated with proliferation; The mRNA levels of *CCNE1*, *CCND1*, *CDK2* and *PCNA* in control and experimental group; **: *p* < 0.01. (**C**) Overexpression of 1 μg overexpression plasmid inhibits goat subcutaneous preadipocytes proliferation by crystal violet staining analysis; the images are representative of the results obtained (100×). All experiments in our study were performed at least in triplicate (**D**) Semi-quantification was used to examine the cell number; *: *p* < 0.05. (**E**) MTT was used to check cell viability; **: *p* < 0.01.

**Figure 7 animals-13-01757-f007:**
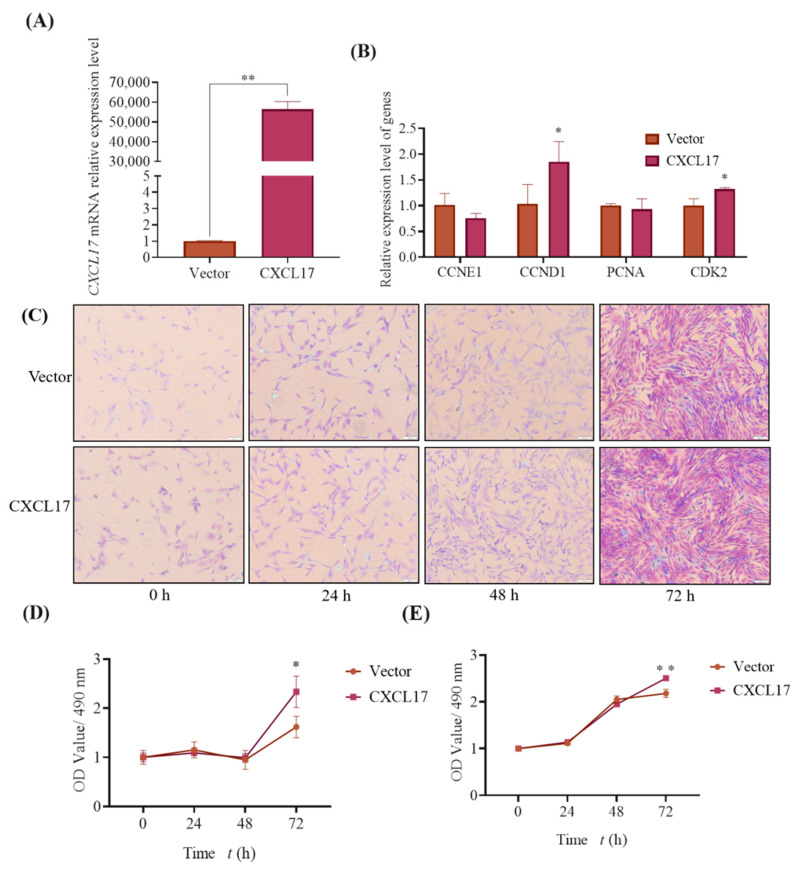
Overexpression of high-dose *CXCL17* promotes goat subcutaneous preadipocytes proliferation. (**A**) Overexpression efficiency of *CXCL17* gene when transfected with 2 μg overexpression plasmid; **: *p* < 0.01. (**B**) Expression of genes associated with proliferation; The mRNA levels of *CCNE1*, *CCND1*, *CDK2* and *PCNA* in control and experimental group, *: *p* < 0.05. (**C**) 2 μg overexpression plasmid inhibits goat subcutaneous preadipocytes proliferation by crystal violet staining analysis; The images are representative of the results obtained (100×). (**D**) Semi-quantification was used to examine the cell number; *: *p* < 0.05. (**E**) MTT was used to check cell viability; **: *p* < 0.01.

**Figure 8 animals-13-01757-f008:**
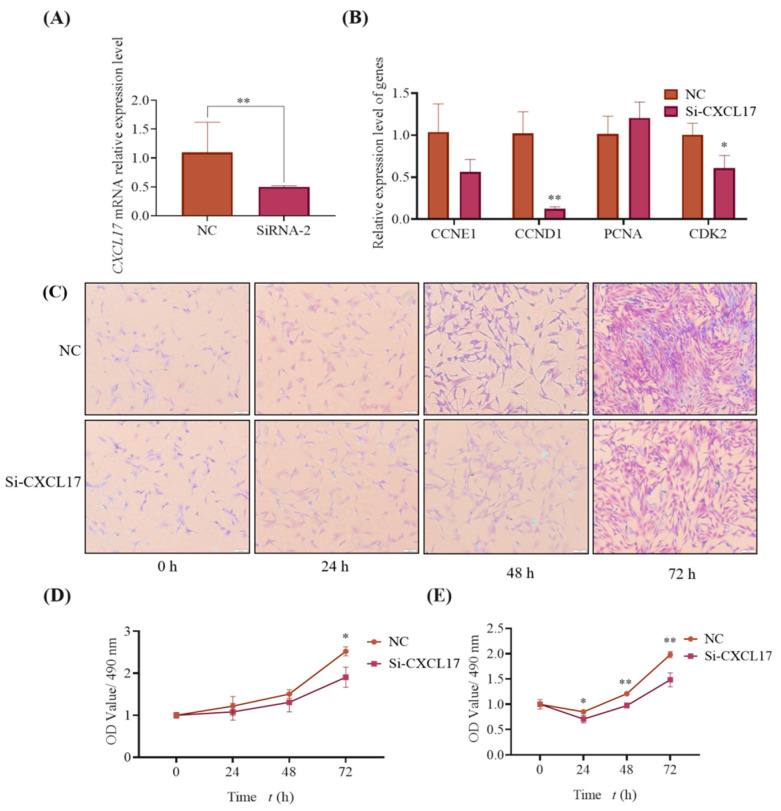
Knockdown of *CXCL17* inhibits goat subcutaneous preadipocytes proliferation. (**A**) The knockdown efficiency of si-*CXCL17* at mRNA; **: *p* < 0.01 (**B**) Expression of genes associated with proliferation; The mRNA levels of *CCNE1*, *CCND1*, *CDK2* and *PCNA* in control and experimental group, **: *p* < 0.01, *: *p* < 0.05. (**C**) knockdown of *CXCL17* inhibits goat subcutaneous preadipocytes proliferation by crystal violet staining analysis; The images are representative of the results obtained (100×). (**D**) Semi-quantification was used to examine the cell number; *: *p* < 0.05. (**E**) MTT was used to check cell viability; **: *p* < 0.01, *: *p* < 0.05.

**Figure 9 animals-13-01757-f009:**
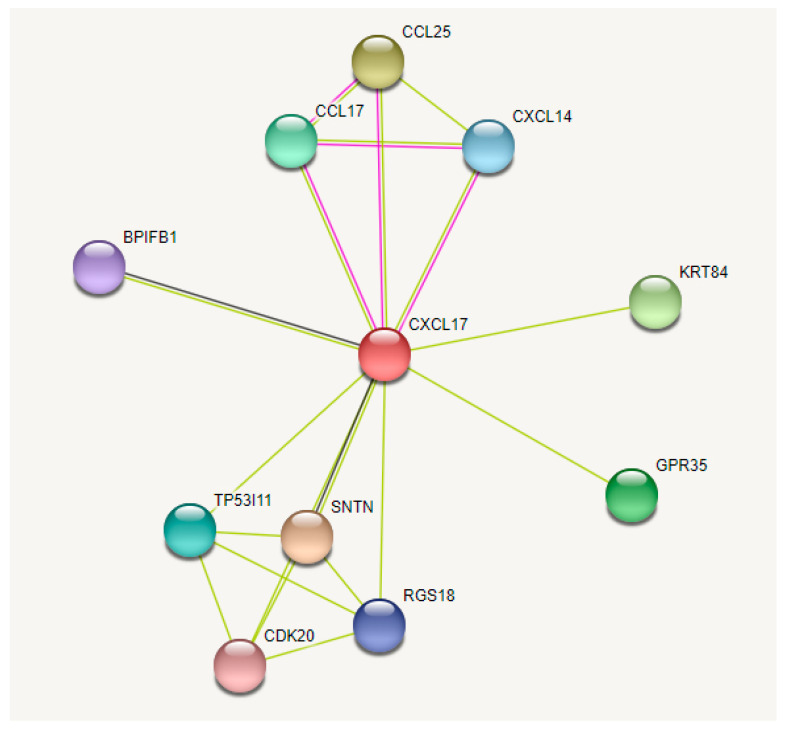
Interaction diagram of goat *CXCL17* protein.

**Table 1 animals-13-01757-t001:** Contents and tools of Bioinformatics analysis.

Analytical Contents	Analytical Software or Online Tools
Nucleotide sequence alignment	DNAMAN
Amino acid sequence translation	ORF Finder (NCBI)
Prediction of phosphorylation sites	NetPhos 3.1
Prediction of signal peptide sites	SignalP-5.0 Server
Prediction of transmembrane domain	TMHMM
Subcellular localization analysis	PSORT Ⅱ
Prediction of secondary structure protein	SOPMA
Prediction of protein tertiary structure	SWISS-MODEL
Protein–protein interaction analysis	STRING
Phylogenetic tree construction	MEGA 5.05
Conserved Domain analysis	Conserved Domain (NCBI)
Physical and chemical properties and primary structure analysis	Ex PASy-ProParam

**Table 2 animals-13-01757-t002:** Primer information.

Primer Names	Primer Sequence (5′→3′) *	Product Length (bp)	Tm (*T*/°C)	Purpose
*CXCL17*	F: 5′ CCTGTTGCTGCCACTAATGC 3′ R: 5′ GTGATGTCTTCGGTGTCTGGT 3′	250	60	qPCR
*CXCL17*	F: 5′ CTAGAATTCTGATGAAGGTTCTAATCTCTTCCC 3′ ^1^ R: 5′ CGGGGTACCACTAAGGGCAGAACAAAGCTT 3′	357	60	plasmid construction
*UXT*	F: 5′ GCAAGTGGATTTGGGCTGTAAC 3′ R: 5′ ATGGAGTCCTTGGTGAGGTTGT 3′	180	60	qPCR
*CCNE1*	F: 5′ CTCCCTGATTCCCACACCTG 3′ R: 5′ CATAAGATGCTTGTCCCTCA3′	193	60	qPCR
*PCNA*	F: 5′ AGTGGAGAACTTGGAAATGGAA 3′ R: 5′ GAGACAGTGGAGTGGCTTTTGT 3′	154	60	qPCR
*CCND1*	F: 5′ TGAACTACCTGGACCGCT 3′ R: 5′ CAGGTTCCACTTGAGTTTGT 3′	212	60	qPCR
*CDK2*	F: 5′ GCCAGGAGTTACTTCTATGC 3′ R: 5′ TGGAAGAAAGGGTGAGCC 3′	180	60	qPCR

* F: Sense primer; R: Antisense primer. ^1^ The double underlined lines are restriction sites, and the single underlined are bases protecting the restriction sites.

## Data Availability

Not applicable.

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
