# Peer review of "Effect of CXCL17 on Subcutaneous Preadipocytes Proliferation in Goats"

_animals, 2023, doi:10.3390/ani13111757_

Round 1

Reviewer 1 Report

1.    References should be included for Table 1 analysis software and online tools.

2.    Section 3.4, The results indicated that CXCL17 may play different regulatory roles at different stages of differentiation of goat subcutaneous preadipocytes. Why not study the effect of CXCL17 on goat preadipocyte differentiation, but study the role of CXCL17 on preadipocyte proliferation? How to distinguish goat preadipocyte proliferation and differentiation?

3.    Using the pEGFP-CXCL17 overexpression plasmid, the authors obtained a ~2,400,000-fold overexpression (Figure 5D), ~4,000-fold overexpression (Figure 6A), ~55,000-fold change (Figure 7A), which is non-physiologic, please explain. Can they obtain significant results by overexpressing CXCL17 by 10-fold or less?

4.    The author only uses two transfection doses (1 μg and 2 μg), 1 μg is defined as a low dose, and 2 μg is defined as a high dose. Is it reasonable?

English language needs to be modified appropriately.

Author Response

Point 1: References should be included for Table 1 analysis software and online tools.

Response 1: Thank you very much for your valuable suggestion. We added the references (reference 26. Sheng XQ, Zhao N, Lin YQ, et al. Cloning and expression analysis of goat ZNF32. Biotechnology Bulletin. 2022,38(12):300-311. doi:10.13560/j.cnki.biotech.bull.1985.2022-0098. ) containing the analysis software and online tools presented in Table 1, and the text has been revised accordingly (line 114).

Point 2: Section 3.4, The results indicated that CXCL17 may play different regulatory roles at different stages of differentiation of goat subcutaneous preadipocytes. Why not study the effect of CXCL17 on goat preadipocyte differentiation, but study the role of CXCL17 on preadipocyte proliferation? How to distinguish goat preadipocyte proliferation and differentiation?

Response 2: Thank you for your valuable comment. CXCL17 is a molecule that our laboratory pays close attention to, and we will explore its effect and mechanism on fat deposition in different parts of goats. Lipid deposition mainly due to the proliferation of preadipocytes and differentiation of adipocytes. At present, we have clarified the effects of CXCL17 on the proliferation of goat subcutaneous adipocytes, and the study on its effect on subcutaneous adipocyte differentiation is in progress, which will be reported in the future. Cell proliferation and differentiation are two different biological processes, which can be detected by their respective markers and biological phenomena. For example, cell proliferation will show increased cell number, and proliferation-related molecules such as CCNE1, CCND1, PCNA, CDK2 will be up-regulated. While cell differentiation will show enlarged lipid droplets, accumulation increased and differentiation related markers such as CEBP-α and CEBP-β were up-regulated.

Point 3: Using the pEGFP-CXCL17 overexpression plasmid, the authors obtained a ~2,400,000-fold overexpression (Figure 5D), ~4,000-fold overexpression (Figure 6A), ~55,000-fold change (Figure 7A), which is non-physiologic, please explain. Can they obtain significant results by overexpressing CXCL17 by 10-fold or less?

Response 3: Thank you very much for your valuable question. Hieff TransTM Liposomal Transfection Reagent instructions recommend a dose of 2-4 μg DNA for transfection. When detecting the overexpression efficiency, we transfected 3 μg and obtained 2,400,000 times of overexpression (Figure 5D), and transfected 2 μg and obtained 55,000 times of overexpression (Figure 6A). Although the multiple of overexpression changed with the amount of transfected expression plasmid, the cell state of expression plasmid at 3 μg was not as good as 2 μg. So we selected 2 μg for subsequent overexpression experiment and took it as the transfection dose of normal plasmid.

Point 4: The author only uses two transfection doses (1 μg and 2 μg), 1 μg is defined as a low dose, and 2 μg is defined as a high dose. Is it reasonable?

Response 4: Thank you for your valuable question. Through the preliminary experimentswe found that 2 μg plasmids were the optimal transfection amount of overexpression experiment for 6-well plate transfection, so we defined that 2 μg was the high dose and 1 μg was the low dose.

Reviewer 2 Report

Dear authors, the topic of your work “Effect of CXCL17 on Subcutaneous Preadipocyte Proliferation in Goats .” it is very scientific importance in goats production.

I would like the following considerations to the authors:

Simple summary 

Line 11- When you use the first time “the role of CXCL17 on adipose deposition…” I recommend use “the role of CXCL17 gen on adipose deposition “

Abstract is ok. 

The introduction is too brief, and I would focus more on the context of the article's subject matter and its applications. 

Material and Methods are very well structured and I have no questions.

The results are clearly presented and are accompanied by a wealth of figures and graphs. 

The discussion of the results is good. 

The conclusions lack a focus on the objectives and an emphasis on their practical application.

In my opinion this is a very good work for publish in Animals correcting these topics.  

Congratulations. 

Author Response

Point 1: Line 11- When you use the first time “the role of CXCL17 on adipose deposition…” I recommend use “the role of CXCL17 gene on adipose deposition “

Response 1: Thank you very much for your valuable comment and suggestion, we have modified the sentence as follows: However, no studies have reported the role of CXCL17 gene on adipose deposition in goats before. And the text has been revised accordingly (line 10 to line 11).

Point 2: Abstract is ok.

Response 2: Thank you very much for your valuable comment.

Point 3: The introduction is too brief, and I would focus more on the context of the article's subject matter and its applications.

Response 3: Thank you very much for your valuable suggestion. We revised the introduction and supplemented the subject and applications. And the text has been revised accordingly (line 40 to line 48, line 72 to line 80, line 87 to line 88).

Point 4: Material and Methods are very well structured and I have no questions.

Response 4: Thank you very much for your valuable comment.

Point 5: The results are clearly presented and are accompanied by a wealth of figures and graphs.

Response 5: Thank you for your valuable comment.

Point 6: The discussion of the results is good.

Response 6: Thank you very much for your valuable comment.

Point 7: The conclusions lack a focus on the objectives and an emphasis on their practical application.

Response 7: Thank you very much for your valuable suggestion, we have supplemented the objectives and applications as follows: These results indicate that CXCL17 is a novel candidate gene for adipose deposition in goats. This study will provide a new theoretical basis for further study of lipid deposition in goat adipocytes and provide a basis for enriching the molecular regulatory network and mechanism of goat CXCL17. And the text has been revised accordingly (line 463 to line 466).

Reviewer 3 Report

I would like to ask some points to the authors.

Please discuss the possibility that the phenomenon observed in this study for CXCL17 is specific to goat or not.

In statistical analysis, which did the authors actually show, SD or SE?

In Figure 4, what alphabets (A, B, C) means?

Please discuss the impact of using fewer samples on the results of this study?

Minor spell check by the authors might improve English quality.

Author Response

Point 1: Please discuss the possibility that the phenomenon observed in this study for CXCL17 is specific to goat or not.

Response 1: Thank you very much for your valuable suggestion. Since our sample came from Jianzhou Daer goats, whether tissue expression detection experiment had the same results in other goat breeds is unknown and needs to be verified. Related studies reported that low-dose recombinant CXCL17 reduced the number of spleen cells in mice, while high-dose recombinant CXCL17 increased the number of spleen cells in mice [19] (reference 19. Matsui A, Yokoo H, Negishi Y, et al. CXCL17 expression by tumor cells recruits CD11b+Gr1 high F4/80- cells and promotes tumor progression. PLoS One. 2012;7(8):e44080. doi:10.1371/journal.pone.0044080.). Different doses of CXCL17 showed different biological effects on the cells which was consistent with the phenomenon of CXCL17 in goat subcutaneous preadipocytes in this study. Therefore, the CXCL17 phenomenon observed in this study may not be unique to goats. And the text has been revised accordingly (line 424 to line 426, line 450 to line 457).

Point 2: In statistical analysis, which did the authors actually show, SD or SE?

Response 2: Thank you very much for your valuable question. In statistical analysis, our data is expressed as "mean ± standard deviation (mean ±SD)". We have modified the “standard error” to the “standard deviation”, and the text has been revised accordingly (line 188).

Point 3: In Figure 4, what alphabets (A, B, C) means?

Response 3: Thank you very much for your valuable question. Different capital letters indicate extremely significant differences (P <0.01, n=3). The same capital letters indicate insignificant differences (P >0.05, n=3). These letters are annotated for statistical analysis, and to present our results more clearly, we replace them with “* *” and “*”. And the text has been revised accordingly (Figure 4).

Point 4: Please discuss the impact of using fewer samples on the results of this study?

Response 4: Thank you very much for your valuable suggestion. We have added “our test results were a replication of three goats, and it is unknown whether increasing the sample size will affect the tissue expression profile of CXCL17 in goats.” to the discussion. And the text has been revised accordingly (line 426 to line 428).

Reviewer 4 Report

Dear Authors,

I attentively studied your work and found it well done from a technical and experimental standpoint, with a certain, albeit not high, level of interest, however I find the data presented in an insufficient manner. I believe that the work can be published but only after a thorough revision. Below you will find my instructions.

Main suggestion

1. You have to indicate from which idea this work was born, considering that I have not found any article (for any species) that relates CXCL17 with adipose tissues.

2. You say nothing about the structure of the gene (4 exons), and this information must be reported

3. The statistical analysis part needs to be expanded, for example: how many replicates for each qPCR did you do?

4. I have a lot of questions regarding how to understand the results, and your discussion hasn't helped me clear them up. To summarize:

a. CXCL17 is expressed in adipose tissue;

b. low-dose CXCL17 overexpression inhibits adipose tissue growth;

c. high-dose CXCL17 overexpression enhances adipose tissue growth; and

d. KO of expression decreases adipose tissue growth.

The difference in effects between the two overexpressions is inexplicable to me and therefore I suggest you clarify your hypothesis in the discussion.

Secondary suggestions.

Line 65.          In this study, we cloned goat CXCL17 coding sequence….. (not gene)

Line 68.          it is necessary to report the primers used and where they are positioned on the bovine sequence.

Line 87.          Cloning of coding region of goat CXCL17

Line 93.          Bioinformatics analysis of goat CXCL17 coding region

Line 107.        Table 2 does not only include the primers used to verify the expression, so it must be reviewed and cited where appropriate.

Table 2.           a. what does it mean that a primer pair was used to check for overexpression? b. for UXT gene report qPCR and not reference. c. Why did you use such large amps for PCR experiments? normally, to get precise results, it is suggested to stay below 89/90 bp.

Line 171.        ….we cloned its coding sequence first.

Line 178.        Amplification of CXCL17 coding region in goat.

Line 179.        PCR products obtained in the amplification of the CXCL17 coding region.

Line 180.        acid of CXCL17 coding region in goat.

Line 181.        there is no TAG triplet in the sequence displayed, but the stop triplet is mnissing….

Figure 2a.       This figure lacks the reference axis for percentages.

Figure 2b.       I am not an expert in the sector, but in my opinion there is no legend of the colors and sizes of the bands indicated.

Line 209.        figure 3a is unnecessary and not suitable for a scientific paper.

Figure 3e/f.     I believe that figure 3e should be deleted and the sequence number and homology data inserted into figure 3f, which would then become 3e. Also why are there 2 bos mutus?

Figura 4.         All graphs related to the expression value must be merged into one graph. The way they are presented makes it difficult to see the differences between the various types of fabrics. It also reported that “Because CXCL17 gene expression was the highest in subcutaneous adipose”, but the value of the expression seems to be at least 10 times lower between lung and subcutaneous adipose…. Finally, the discussion of these results needs to be reviewed considering the new graph. Furthermore, in graph 4 a,b,c and d (which will become a new unic graph) the asterisks relating to statistical significance are not indicated (but cited in the text). Figure 4d to 4h should be placed in a single figure and the statistical analysis adapted to the new format. The way these results are presented is absolutely incorrect!!

Figure 5.         Unfortunately figure 5c, as it is presented, is absolutely incomprehensible and cannot be used to interpret the results.... moreover, the legend is absolutely insufficient!

Line 330.        “To further confirm the inhibitory effect of CXCL17 on goat subcutaneous”: but the inhibitory effect is only for the low dose and not for the high dose...

Line 380.        …..primers and clone goiat coding sequence.

Line 405-37.        This part of the discussion is a repetition of the results and there is little of a real discussion in its own right. I suggest that you write less and include a real discussion

Author Response

Main suggestion

Point 1: You have to indicate from which idea this work was born, considering that I have not found any article (for any species) that relates CXCL17 with adipose tissues.

Response 1: Thank you very much for your valuable question. Since CXCL17 has previously been reported to be involved in cell proliferation, and preadipocyte proliferation promotes lipid accumulation. Goat lipid deposition plays an important role in meat quality improvement. Tissue expression detection result showed that CXCL17 expression level was the highest in subcutaneous adipose tissue. Therefore, we speculated that it plays an important regulatory role in the process of subcutaneous fat deposition in goats. And the text has been revised accordingly (line 72 to line 77).

Point 2: You say nothing about the structure of the gene (4 exons), and this information must be reported

Response 2: Thank you very much for your valuable comment and suggestion, we have added “CXCL17 gene localized on human chromosome 19q13.2 was found in 2006, and consisted of four exons” to the introduction. And the text has been revised accordingly (line 57 to line 59).

Point 3: The statistical analysis part needs to be expanded, for example: how many replicates for each qPCR did you do?

Response 3: Thank you very much for your valuable suggestion, we have added “All experiments in this study including biological information analysis, qPCR, cell culture, MTT, and semi-quantitative experiments were performed in triplicate.” to the 2.10. And the text has been revised accordingly (line 190 to line 192).

Point 4: I have a lot of questions regarding how to understand the results, and your discussion hasn't helped me clear them up. To summarize:

  1. CXCL17 is expressed in adipose tissue;
  2. low-dose CXCL17 overexpression inhibits adipose tissue growth;
  3. high-dose CXCL17 overexpression enhances adipose tissue growth; and
  4. KO of expression decreases adipose tissue growth.

Response 4: Thank you very much for your valuable comment and questions. CXCL17 was expressed in all tissues tested, and the expression level is highest in subcutaneous adipose. Therefore, it was speculated that CXCL17 plays an important regulatory role in the process of subcutaneous fat deposition in goats. And the text has been revised accordingly (line 423 to line 426).

When the expression plasmid was transfected at a dose of 1 μg, CXCL17 expression was knocked out, the expression of proliferation marker genes was decreased, the number of cells was smaller and the cell viability was lower than that of the control group. When the transfection dose of expression plasmid was 2 μg, the opposite result was found. These results indicated that different doses of CXCL17 had different effects on the proliferation of subcutaneous preadipocytes of goats. Related studies reported that low-dose recombinant CXCL17 reduced the number of spleen cells in mice, while high-dose recombinant CXCL17 increased the number of spleen cells in mice[19] (reference 19. Matsui A, Yokoo H, Negishi Y, et al. CXCL17 expression by tumor cells recruits CD11b+Gr1 high F4/80- cells and promotes tumor progression. PLoS One. 2012;7(8):e44080. doi:10.1371/journal.pone.0044080.). Different doses of CXCL17 showed different biological effects on the cells which was consistent with the phenomenon of CXCL17 in goat subcutaneous preadipocytes in this study. Therefore, the CXCL17 phenomenon observed in this study may not be unique to goats. And the text has been revised accordingly (line 450 to line 457).

Point 5: The difference in effects between the two overexpressions is inexplicable to me and therefore I suggest you clarify your hypothesis in the discussion.

Response 5: Thank you very much for your valuable suggestion. The expression vector transfected with 1 μg inhibited the proliferation of goat subcutaneous preadipocytes, while the expression vector transfected with 2 μg promoted the proliferation of goat subcutaneous preadipocytes, and interfered with CXCL17 expression inhibited the proliferation of goat subcutaneous preadipocytes. These results indicated that different doses of CXCL17 had different effects on the proliferation of subcutaneous preadipocytes of goats. Related studies reported that low-dose recombinant CXCL17 reduced the number of spleen cells in mice, while high-dose recombinant CXCL17 increased the number of spleen cells in mice[19] (reference 19. Matsui A, Yokoo H, Negishi Y, et al. CXCL17 expression by tumor cells recruits CD11b+Gr1 high F4/80- cells and promotes tumor progression. PLoS One. 2012;7(8):e44080. doi:10.1371/journal.pone.0044080.). Different doses of CXCL17 showed different biological effects on the cells which was consistent with the phenomenon of CXCL17 in goat subcutaneous preadipocytes in this study. Therefore, the CXCL17 phenomenon observed in this study may not be unique to goats. And the text has been revised accordingly (line 445 to line 457).

Secondary suggestions.

Point 6: Line 65. In this study, we cloned goat CXCL17 coding sequence….. (not gene)

Response 6: Thank you very much for your valuable suggestion, , we have modified Sentence to “In this study, we cloned CXCL17 coding sequence and analyzed its biological characteristics by the bioinformatics methods”. And the text has been revised accordingly (line 81).

Point 7: Line 68. it is necessary to report the primers used and where they are positioned on the bovine sequence.

Response 7: Thank you very much for your valuable suggestion, only the cloned primers of CXCL17 gene in this study were based on the mRNA sequence of cattle, and the other primers were designed based on the CXCL17 sequence cloned. We have supplemented the primer information for CXCL17 gene cloning in the method part. And the text has been revised accordingly (line 106 to line 108).

Point 8: Line 87. Cloning of coding region of goat CXCL17

Response 8: Thank you very much for your valuable suggestion, we have modified “Gene cloning of goat CXCL17” to “Cloning of coding region of goat CXCL17”. And the text has been revised accordingly (line 104).

Point 9: Line 93. Bioinformatics analysis of goat CXCL17 coding region

Response 9: Thank you very much for your valuable suggestion, we have modified “Bioinformatics analysis of goat CXCL17 gene” to “Bioinformatics analysis of goat CXCL17 coding region”. And the text has been revised accordingly (line 112).

Point 10: Line 107. Table 2 does not only include the primers used to verify the expression, so it must be reviewed and cited where appropriate.

Response 10: Thank you very much for your valuable suggestion. We deleted the cloning primer information in Table 2 and added it to 2.3. And information about primers used to construct expression vectors is referenced in 3.5. And the text has been revised accordingly (line 106 to line 108, line 287 to line 288 ).

Point 11: Table 2. a. what does it mean that a primer pair was used to check for overexpression? b. for UXT gene report qPCR and not reference. c. Why did you use such large amps for PCR experiments? normally, to get precise results, it is suggested to stay below 89/90 bp.

Response 11: Thank you very much for your valuable question.

  1. A pair primer was not used to check for overexpression, it was used to construct the CXCL17 overexpression plasmid. We have modified “over expression” to “plasmid construction” in the purpose of Table 2.And the text has been revised accordingly (Table 2).
  2. With your valuable advice, we have modified the purpose of UXTgene in Table 2. And we have added “According to the instructions of SYBR ® Premix Ex Taq TM (2×) kit (TaKara, Japan), the expression levels of CXCL17 and proliferation marker genes were determined using UXT as an internal reference gene. CXCL17 quantitative primers and expression vector construction primers were designed by cloned CXCL17 gene sequence, and other primers were designed by goat sequences in NCBI. And primer information was listed in Table 2.” to 2.6. And the text has been revised accordingly (line 124 to line 128, Table 2).
  3. The size in Table 2 is the product length rather than the primer size. To better present our results,we have modified “Size” to “Product length” in Table 2. And the text has been revised accordingly (Table 2).

Point 12: Line 171.  ….we cloned its coding sequence first.

Response 12: Thank you very much for your valuable suggestion, we have modified “we cloned its gene sequence first” to “we cloned its coding sequence first”. And the text has been revised accordingly (line 197 to line 198).

Point 13: Line 178. Amplification of CXCL17 coding region in goat.

Response 13: Thank you very much for your valuable suggestion, we have modified “Amplification of CXCL17 gene in goat” to “Amplification of CXCL17 coding region in goat”. And the text has been revised accordingly (line 204).

Point 14: Line 179. PCR products obtained in the amplification of the CXCL17 coding region.

Response 14: Thank you very much for your valuable suggestion, we have modified “The CXCL17 gene in goat” to “PCR products obtained in the amplification of the CXCL17 coding region”. And the text has been revised accordingly (line 205).

Point 15: Line 180. acid of CXCL17 coding region in goat.

Response 15: Thank you very much for your valuable suggestion, we have modified “The sequences of open reading frame and deduced amino acid of CXCL17 gene in goat” to “The sequences of open reading frame and deduced amino acid of CXCL17 coding region in goat”. And the text has been revised accordingly (line 206).

Point 16: Line 181. there is no TAG triplet in the sequence displayed, but the stop triplet is mnissing….

Response 16: Thank you very much for your valuable question. The TTA is actually show. TAG occurs in the text because misspelling TTA as TAG. We have modified “TAG” to “TTA”. And the text has been revised accordingly (line 208).

Point 17: Figure 2a. This figure lacks the reference axis for percentages.

Response 17: Thank you very much for your valuable suggestion, we have added the reference axis for percentages to the Figure 2a. And the text has been revised accordingly (Figure 2a).

Point 18: Figure 2b. I am not an expert in the sector, but in my opinion there is no legend of the colors and sizes of the bands indicated.

Response 18: Thank you very much for your valuable suggestion. The SOPMA software was used to predict the secondary structure of CXCL17 in goats. The results showed that 57 amino acids (48.31%) had the highest proportion of random curling, 46 amino acids (38.98%) could form α helix, 12 amino acids could form extension chain (10.17%) and 3 amino acids (2.54%) could form β turn. In Figure 2b, blue represents alpha spirals, red represents extended connections, green represents beta folds, and purple represents random curls. The size of the bands in Figure 2b has no special meaning, just to show the bands more clearly. We have added the legend of the colors of the bands indicated to the Figure 2b. And the Figure 2b has been revised accordingly (Figure 2b).

Point 19: Line 209. figure 3a is unnecessary and not suitable for a scientific paper.

Response 19: Thank you very much for your valuable suggestion, we have deleted the figure 3a and reordered the pictures in Figure 3. And the text has been revised accordingly (Figure 3).

Point 20: Figure 3e/f.  I believe that figure 3e should be deleted and the sequence number and homology data inserted into figure 3f, which would then become 3e. Also why are there 2 bos mutus?

Response 20: Thank you very much for your valuable suggestion. 3e shows some species with high amino acid homology with goats and their related information, while 3f shows the evolutionary distance between species with high amino acid homology with goats. There are two bos mutus are due to spelling errors. Under your reminder, we have modified it. And the text has been revised accordingly (Figure 3).

Point 21: Figure 4. All graphs related to the expression value must be merged into one graph. The way they are presented makes it difficult to see the differences between the various types of fabrics. It also reported that “Because CXCL17 gene expression was the highest in subcutaneous adipose”, but the value of the expression seems to be at least 10 times lower between lung and subcutaneous adipose…. Finally, the discussion of these results needs to be reviewed considering the new graph. Furthermore, in graph 4 a,b,c and d (which will become a new unic graph) the asterisks relating to statistical significance are not indicated (but cited in the text). Figure 4d to 4h should be placed in a single figure and the statistical analysis adapted to the new format. The way these results are presented is absolutely incorrect!!

Response 21: Thank you very much for your valuable suggestion, we have combined 14 tissues on a single expression profile, and placed 4d to 4h  in a single figure. Meanwhile, the statistical analysis adapted to the new format. And the text has been revised accordingly (Figure 4, line 281 to line 282).

Point 22: Figure 5. Unfortunately figure 5c, as it is presented, is absolutely incomprehensible and cannot be used to interpret the results.... moreover, the legend is absolutely insufficient!

Response 22: Thank you very much for your valuable suggestion, we added “In contrast to pEGFP-N1, pEGFP-CXCL17 has many dispersed extracellular fluorescence, largely consistent with the subcellular localization results in 3.3” to 3.5 and “Fluorescence localization revealed that transfected pEGFP-CXCL17 had many extracellular fluorescence, consistent with the CXCL17 precursor protein as a secreted protein [11]” to the discussion. Meanwhile, we modified the original legend to “Fluorescence localization of CXCL17 protein in goat precursor adipocytes cultured in vitro”. And the text has been revised accordingly (line 291 to line 293, line 302, line 409 to line 411).

Point 23: Line 330. “To further confirm the inhibitory effect of CXCL17 on goat subcutaneous”: but the inhibitory effect is only for the low dose and not for the high dose...

Response 23: Thank you very much for your valuable suggestion, we have modified “To further confirm the inhibitory effect of CXCL17 on goat subcutaneous” to the “To further confirm the effect of CXCL17 on goat subcutaneous”. And the text has been revised accordingly (line 357).

Point 24: Line 380.  …..primers and clone goat coding sequence.

Response 24: Thank you very much for your valuable suggestion, we have modified “In this study, the mRNA sequence of cattle CXCL17 gene was used as a template to design primers and clone gene sequence” to “In this study, the mRNA sequence of cattle CXCL17 gene was used as a template to design primers and clone goat coding sequence.”. And the text has been revised accordingly (line 409).

Point 25: Line 405-37.  This part of the discussion is a repetition of the results and there is little of a real discussion in its own right. I suggest that you write less and include a real discussion

Response 25: Thank you very much for your valuable suggestion, we have revised the discussion according to your suggestion. We discussed “The expression vector transfected with 1 μg inhibited the proliferation of goat subcutaneous preadipocytes, while the expression vector transfected with 2 μg promoted the proliferation of goat subcutaneous preadipocytes, and interfered with CXCL17 expression inhibited the proliferation of goat subcutaneous preadipocytes. These results indicated that different doses of CXCL17 had different effects on the proliferation of subcutaneous preadipocytes of goats. Related studies reported that low-dose recombinant CXCL17 reduced the number of spleen cells in mice, while high-dose recombinant CXCL17 increased the number of spleen cells in mice [19]. Different doses of CXCL17 showed different biological effects on the cells which was consistent with the phenomenon of CXCL17 in goat subcutaneous preadipocytes in this study. Therefore, it is highly likely that CXCL17 needs to overcome the threshold before its inhibitory effect on the proliferation of subcutaneous preadipocytes of goats can be converted into promoting effect.” in the discussion. And the text has been revised accordingly (line 445 to line 457).

Round 2

Reviewer 1 Report

No comments

Author Response

Thank you very much for your valuable comment.

Reviewer 3 Report

I have now simple but great concern about the shortage of sample size (n=3), which seems hard to obtain a general conclusion with valid reproducibility.

Short communication might be better choice to publish the results.

Minor spell check would be warranted.

Author Response

Point 1: I have now simple but great concern about the shortage of sample size (n=3), which seems hard to obtain a general conclusion with valid reproducibility.

Response 1: Thank you very much for your valuable comment. We set the number of sample replicates for this experiment according to the relevant literature (Yao Jingjie, Xu Ping, Chen Linghui, et al. Cloning, biological characteristics and mRNA expression analysis of CCL28 gene from cattle. Heilongjiang animal husbandry and veterinary. 2021,No.621(09):12-16+161-162. doi:10.13881/j.cnki.hljxmsy.2020.07.0453.   Gao Xiaoqian, Lei Zhaoxiong, Tang Lin, et al. Cloning and expression profiling of CDS region of bovine HoxC9 gene. Journal of Agricultural University of Hebei. 2021,44(03):73-78. doi:10.13320/j.cnki.jauh.2021.0050.   Du Y, Wang Y, Xu Q, et al. TMT-based quantitative proteomics analysis reveals the key proteins related with the differentiation process of goat intramuscular adipocytes. BMC Genomics. 2021;22(1):417. Published 2021 Jun 5. doi:10.1186/s12864-021-07730-y. ). So, the conclusion has certain valid reproducibility.

Point 2: Short communication might be better choice to publish the results.

Response 2: Thank you very much for your valuable comment and suggestion. The content of this research manuscript is more in line with the requirements of the article that reports scientifically sound experiments and provides a wealth of new information than the requirements of the newsletter in Animals.

Point 3: Minor spell check would be warranted.

Response 3: Thank you very much for your valuable comment and suggestion. We have checked the full text for spelling and corrected the spelling mistakes.

Reviewer 4 Report

Dear Authors,

I have read your revided version carefully and am completely satisfied with the changes you have made.

For this reason I have suggested publication in this form.

Author Response

(The authors gave the same response as above.)
